# The Effect of Preoperative Administration of Glucocorticoids on the Postoperative Complication Rate in Liver Surgery: A Systematic Review and Meta-Analysis of Randomized Controlled Trials

**DOI:** 10.3390/jcm13072097

**Published:** 2024-04-03

**Authors:** Caner Turan, Emőke Henrietta Kovács, László Szabó, Işıl Atakan, Fanni Dembrovszky, Klementina Ocskay, Szilárd Váncsa, Péter Hegyi, László Zubek, Zsolt Molnár

**Affiliations:** 1Centre for Translational Medicine, Semmelweis University, 1085 Budapest, Hungary; c.caner.turan@gmail.com (C.T.); islatakan@gmail.com (I.A.); dembrovszky.f@gmail.com (F.D.); hegyi2009@gmail.com (P.H.);; 2Department of Anesthesiology and Intensive Therapy, Semmelweis University, 1085 Budapest, Hungary; 3Selye János Doctoral College for Advanced Studies, Semmelweis University, 1085 Budapest, Hungary; 4Institute for Translational Medicine, Medical School, University of Pécs, 7623 Pécs, Hungary; 5Institute of Pancreatic Diseases, Semmelweis University, 1085 Budapest, Hungary; 6Department of Anaesthesiology and Intensive Therapy, Poznan University of Medical Sciences, 61-701 Poznan, Poland

**Keywords:** glucocorticoid, liver surgery, perioperative mangement

## Abstract

**Background**: Glucocorticoids may grant a protective effect against postoperative complications. The evidence on their efficacy, however, has been inconclusive thus far. We investigated the effects of preoperatively administered glucocorticoids on the overall postoperative complication rate, and on liver function recovery in patients undergoing major liver surgery. **Methods:** We performed a systematic literature search on PubMed, Embase, and CENTRAL in October 2021, and repeated the search in April 2023. Pre-study protocol was registered on PROSPERO (ID: CRD42021284559). Studies investigating patients undergoing liver resections or transplantation who were administered glucocorticoids preoperatively and reported postoperative complications were eligible. Meta-analyses were performed using META and DMETAR packages in R with a random effects model. Risk of bias was assessed using RoB2. **Results:** The selection yielded 11 eligible randomized controlled trials (RCTs) with 964 patients. Data from nine RCTs (*n* = 837) revealed a tendency toward a lower overall complication rate with glucocorticoid administration (odds ratio: 0.71; 95% confidence interval: 0.38–1.31, *p* = 0.23), but it was not statistically significant. Data pooled from seven RCTs showed a significant reduction in wound infections with glucocorticoid administration [odds ratio: 0.64; 95% confidence interval: 0.45–0.92 *p* = 0.02]. Due to limited data availability, meta-analysis of liver function recovery parameters was not possible. **Conclusions**: The preoperative administration of glucocorticoids did not significantly reduce the overall postoperative complication rate. Future clinical trials should investigate homogenous patient populations with a specific focus on postoperative liver recovery.

## 1. Introduction

Despite advancements in surgical techniques, liver surgery remains a relatively high-risk procedure, with complication rates reaching up to 48% [1]. The most common complications of liver resections and transplantations include postoperative collections, sepsis, and wound and organ space infections. Underlying the complications are thought to be hepatocellular injury and subsequent inflammation, the accumulation of toxic metabolites due to hepatic dysfunction, and a predisposition to coagulopathy and infections [2,3].

Aside from their other effects, hydrocortisone and methylprednisolone, which are both glucocorticoids, have been investigated in the past in both human and animal models for their anti-inflammatory properties, which could be helpful in reducing the postoperative hyperinflammatory state [4,5,6]. Preoperative glucocorticoid administration, based on this pharmacological basis, has been investigated in multiple fields of surgery for its effect on reducing postoperative complication rates [7,8,9,10]. However, the efficacy of routine glucocorticoid administration remains controversial. 

Clinical trials on preoperative glucocorticoid administration in liver surgery have been ongoing since 1996. The 2016 Enhanced Recovery After Surgery guideline on liver surgery recommends glucocorticoids, albeit with a moderate level of recommendation on a weak level of evidence [11]. This guideline references two systematic reviews by Richardson et al. [12] and Li et al. [13], which contradict each other in their results on postoperative complication rates. Since then, two additional systematic reviews have been published on the subject, in 2019 and 2021, by Yang et al. [14] and Hai et al. [15], respectively. However, these two papers also reported contradicting results.

Therefore, we decided to perform a systematic review and meta-analysis to update the current knowledge on the subject. We aimed to summarize and contextualize the existing evidence, based on two hypotheses: (1) preoperative glucocorticoid administration can reduce the complication rate following any type of liver surgery; (2) the effect of glucocorticoids on some complications will be different than on the overall complication rate. Therefore, we investigated not only the overall postoperative complication rate but also distinct complications and liver function parameters, to inform future clinical research and critically appraise the current level of evidence certainty. 

## 2. Methods

We reported our systematic review and meta-analysis in accordance with the PRISMA 2020 Statement [16] (Appendix A: PRISMA 2020 Checklist), and we undertook our research based on the recommendations of the Cochrane Handbook for Systematic Reviews of Interventions [17]. The study protocol was registered on PROSPERO (registration number: CRD42021284559). However, we deviated from the registered protocol concerning reporting our primary outcome, the overall postoperative complication rate. We had initially aimed to report complications following the Clavien–Dindo Classification System [18]. However, this was not possible due to inadequate data availability.

### 2.1. Search Strategy

Our systematic search was conducted on 15 October 2021, on MEDLINE via PubMed, Embase, and the Cochrane Central Register of Controlled Trials (CENTRAL) databases, with no filters and no restrictions on date of publication, language, or article type. This systematic search was repeated on 1 April 2023 to detect any new literature eligible for inclusion. During the systematic search, the following search key was used: (((hepatic OR liver) AND (surgery OR resection OR operation OR intervention)) OR hepatectomy) AND (steroid OR corticosteroid OR glucocorticoid OR methylprednisolone OR hydrocortisone OR cortisol) AND random*. A modified search key was used for the search on Embase: ((hepatic OR ‘liver’/exp OR liver) AND (‘surgery’/exp OR surgery OR ‘resection’/exp OR resection OR ‘operation’/exp OR operation OR ‘intervention’/exp OR intervention) OR ‘hepatectomy’/exp OR hepatectomy) AND (‘steroid’/exp OR steroid OR ‘corticosteroid’/exp OR corticosteroid OR ‘glucocorticoid’/exp OR glucocorticoid OR ‘methylprednisolone’/exp OR methylprednisolone OR ‘hydrocortisone’/exp OR hydrocortisone OR ‘cortisol’/exp OR cortisol) AND random*. References from the selected articles were also searched for additional studies to be included in the selection process.

### 2.2. Eligibility Criteria

Only randomized controlled trials (RCTs) published in peer-reviewed journals and investigating the preoperative administration of glucocorticoids (natural or synthetic) against placebo or non-administration for patients undergoing liver surgery were included in this study. We report the study framework and eligibility criteria according to the PICOS method, where population (P): adult patients (aged 18 or older) of both sexes undergoing elective or non-elective liver surgery, including open or laparoscopic resection or liver transplantation; intervention (I): preoperatively administered high-dose glucocorticoids as a study drug, regardless of dosing strategy, as opposed to standard of care; control (C): placebo or non-administration; main outcome (O): overall postoperative complication rate (referring to the number of patients who experienced any postoperative complication related to the surgical procedure, including but not limited to infections, bile leakage, liver failure, bleeding, and pleural effusion); and setting (S): perioperative hospital care. We included studies that fit the inclusion criteria regardless of the preoperative dosage strategy. Exclusion criteria were study designs other than RCTs, animal studies, and patients who underwent surgeries that included organs other than the liver. Studies were considered eligible for synthesis if they satisfied the eligibility criteria and reported raw data for any or all outcomes under investigation as per our pre-registered study protocol. Publications in which the study population may have overlapped with an earlier publication were not eligible for inclusion. 

### 2.3. Selection Process 

The selection was performed by two teams of independent review authors (CT as review author 1, and IA and EHK as review author 2). Duplicates were detected and removed by both manual and automatic searches. The two reviewer groups then assessed the results for inclusion, first by title and abstract selection, then by full-text selection using EndNote 20 software (Clarivate Analytics, Philadelphia, PA, USA). As agreed, any conflict was resolved by a third independent investigator (FD). To evaluate inter-reviewer agreement, Cohen’s Kappa was calculated once after title-and-abstract selection and once after full-text selection, with κ = 0.97 and κ = 1.0, respectively. Regarding studies with identical patient populations, the reviewers chose to include only the article with the earlier publication date. 

### 2.4. Data Collection Process

From the eligible articles, data were collected by three authors (CT, IA, and EHK) independently. Disagreements were solved by discussion between the authors. The following data were extracted: (1) study characteristics: first author, the year of publication, study design, study population (number, age, and sex), study period, study country, and institute; (2) postoperative complications: overall postoperative complication rate, wound infection, septic/infectious complications, bile leakage, pleural effusion, gastrointestinal bleeding, intra-abdominal bleeding, high-grade liver failure, and all grades of liver failure; (3) laboratory outcomes (total bilirubin, alanine aminotransferase (ALT), aspartate aminotransferase (AST), interleukin-6 (IL-6), C-reactive protein (CRP), and prothrombin time–international normalized ratio (PTT)); (4) perioperative outcomes (length of hospital stay, total operative time, intraoperative blood loss, blood transfusions, and blood products used (FFP or RBC).

When unavailable in writing, data estimates from visual sources were collected using software GetData Graph Digitizer version number: v.2.26), although these estimates were not used in the quantitative synthesis.

### 2.5. Study Risk of Bias and Certainty of Evidence Assessment

Two authors (CT, IA) performed the risk of bias assessment independently, according to the recommendations of the Cochrane Handbook [17], utilizing the RoB 2 tool (ROB2 IRPG beta v6, 25 June 2019) based on the RoB 2 version dated 15 March 2019 [19]. Disagreements were solved by deliberation between the authors. The risk of bias was thus assessed on five distinct domains, including the randomization process, deviations from intended intervention, missing outcome data, the measurement of the outcome, the selection of the reported outcome, and overall bias. The level of certainty of evidence evaluation, using the GRADE assessment based on the GRADE handbook [20], was made using the online software GRADE Pro GDT version 20 [21].

### 2.6. Statistical Analysis

Meta-analysis was performed for outcomes for which at least 3 distinct included studies reported data. The statistical analyses were made using R (R Core Team 2021, v4.1.1) [22]. For calculations and plots, we used the META (Schwarzer 2022, v5.2.0) [23] and DMETAR (Cuijpers, Furukawa, and Ebert 2022, v0.0.9000) [24] packages.

For dichotomous outcomes, the odds ratio (OR) with a 95% confidence interval (CI) was used to measure the effect. To calculate the odds ratio, the total number of patients in each group and those with the event of interest were extracted from each study. Raw data from the selected studies were pooled using a random effects model via the Mantel–Haenszel method (Mantel and Haenszel 1959; Robins, Greenland, and Breslow 1986; Thompson, Turner, and Warn 2001) [25,26,27]. If the study number for the given outcome was over five, the Hartung–Knapp adjustment (Knapp and Hartung 2003; IntHout, Ioannidis, and Borm 2014) [28,29] was applied (below six studies, no adjustment was applied). For the pooled results, an exact Mantel–Haenszel method (no continuity correction) was used to handle zero-cell counts (Cooper, Hedges, and Valentine 2009; J. Sweeting, J. Sutton, and C. Lambert 2004) [30,31]. In individual studies, the zero-cell-count problem was adjusted using treatment arm continuity correction. To estimate τ2, we used the Paule–Mandel method (Paule and Mandel 1982) [32], and the Q-profile method for calculating the confidence interval of τ2 (Harrer et al., 2021) [33]. Statistical heterogeneity across trials was assessed by means of the Cochrane Q test and the I2 values (Higgins and Thompson 2002) [34]. Raw data were used in all instances; in the case of binary data, numbers of event and non-event and, in the case of continuous data, mean and standard deviation (SD) were used. If the mean and SD were not reported in the article, estimations were made using the given values of medians, quartiles, minimums, and maximums, using the Luo [35] and Shi [36] methods.

Forest plots (Rücker and Schwarzer 2021; IntHout et al., 2016) [37,38] were used to graphically summarize the results. 

Outlier and influence analyses were carried out following the recommendations of Harrer et al. (2021) [33] and Viechtbauer and Cheung (2010) [39].

## 3. Results

### 3.1. Study Selection and Characteristics

The systematic search yielded 8226 records, and the selection process is detailed in the flowchart according to PRISMA as presented in Figure 1. Overall, 11 articles [40,41,42,43,44,45,46,47,48,49,50] were included in our study. The repeat search did not find any further studies eligible for inclusion.

### 3.2. Main Characteristics of the Included Studies

In summary, we managed to analyze data from 964 patients, of whom 477 were in the glucocorticoid group, and 487 in the control group. Baseline characteristics, clinical data, and intervention summaries of the included articles are detailed further in Table 1. 

### 3.3. Postoperative Complications

Nine [43,44,45,46,47,48,49,50,51] (*n* = 836) out of the eleven eligible articles in our study reported the overall postoperative complication rate as an outcome. This outcome did not distinguish between major and minor complications or different pathomechanisms. In this analysis, 418 patients were in the intervention group and received glucocorticoids preoperatively, and 419 patients in the control group received either saline or a placebo or nothing. There was a tendency toward a lower overall postoperative complication rate in the intervention group (OR:0.71; 95% CI: 0.38–1.31, *p* = 0.23), but the result did not reach statistical significance (see Figure 2A). There was substantial heterogeneity as defined by the Cochrane Handbook [17] [I^2^ = 54% (2%; 78%), *p* = 0.03].

Five studies [40,41,42,45,47] (*n* = 651) reported the rate of pleural effusion as an outcome. Our analysis found no statistically significant difference between the groups with a tendency toward a lower rate in the intervention group (OR: 0.81; 95% CI: 0.44–1.48, *p* = 0.4963) (see Figure 2B). Seven studies [40,41,42,45,46,48,49] (*n* = 745) reported the rate of wound infection as an outcome. Our analysis found that the intervention significantly reduced wound infections (OR = 0.64; 95% CI: 0.45–0.92, *p* = 0.0241) (see Figure 2C). Four studies [40,41,42,45] (*n* = 598) reported septic/infectious complications as an outcome. Our analysis found no statistically significant difference between the groups with a tendency toward a lower rate in the intervention group (OR: 0.73; 95% CI: 0.24–2.20, *p* = 0.577) (see Figure 2D). Seven studies [40,41,42,45,46,48,49] (*n* = 745) reported the rate of bile leakage as an outcome. Our analysis found no statistically significant difference between the groups (OR: 1.12; 95% CI: 0.59–2.13, *p* = 0.7263) with a tendency toward a higher rate in the intervention group (see Figure 2E). Five studies [40,42,45,46,48] (*n* = 551) reported liver failure as an outcome. Our analysis found no statistically significant difference between the groups (OR: 0.96; 95% CI = 0.49–1.88, *p* = 0.9034) (see Figure 2F).

### 3.4. Laboratory Outcomes

Due to the discrepancy in the methodology of measurements and the reporting of the laboratory outcomes between the included studies, we could not perform a meta-analysis for these parameters. Hence, we included these only in the systematic review. Nevertheless, several individual studies reported statistically significant results. A detailed summary of the measurement time points, results and, where available, *p*-values of each included study are depicted in Appendix A. 

### 3.5. Other Outcomes

Our analysis also included perioperative outcomes. There were no statistically significant differences between the glucocorticoid and control groups with respect to these outcomes. Eight studies [43,44,46,47,48,49,50,52] (*n* = 759) reported on the length of hospital stay (days), (MD: −0.12; 95% CI: −0.57–0.34) (see Figure 3A). Seven studies [43,44,46,47,48,49,52] (*n* = 709) reported on the total operative time (minutes), (MD: −2.82; 95% CI = −19.46–13.83) (see Figure 3B). Eight studies [43,44,45,46,47,48,49,52] (*n* = 857) reported on the blood loss (milliliters), (MD = 3.41; 95% CI: −33.33–40.16) (see Figure 3C). Five studies (*n* = 572) reported on the number of patients who needed to be administered blood transfusion intraoperatively, (OR: 1.04; 95% CI = 0.63–1.71, *p* = 0.89) (see Figure 3D).

### 3.6. Risk of Bias and Study Heterogeneity Assessment

The results of the risk of bias assessment for the outcomes are presented in Figure 4. All outcomes meta-analyzed in this paper received the same score; therefore, Figure 4 represents the results of the assessments of all outcomes.

Overall, most of the included studies were adequately randomized, and no studies had issues arising from missing outcomes. The main risk of bias was related to the inadequate elaboration of the study designs in some cases, which led to some concerns and, in other cases, bias arising from the reporting of the outcomes represented a critical risk.

Levels of heterogeneity are interpreted according to the Cochrane Handbook [17] using τ2, I^2^, and Cochrane Q test statistics [32,33,34]. Moderate heterogeneity (I^2^ = 54% [2%;78%], *p* = 0.03) was observed in the analysis of the overall postoperative complication rate. This may be due to the fact that fewer than ten studies were included in the analysis, and the fact that patients who underwent different liver surgeries were pooled together. Moderate heterogeneity was observed in the analyses of the length of hospital stay (I^2^ = 38% [0%;73%], *p* = 0.12) and blood loss (I^2^ = 40% [0%;73%], *p* = 0.11), possibly due to the difference in the surgical characteristics of the included patients. Severe heterogeneity (I^2^ = 65%, [0%;88%], *p* = 0.03) was observed in the analysis of septic/infectious complications. This could be explained by the size of the patient pool (*n* = 200), given that this analysis only incorporated four studies. No severe heterogeneity has been detected in any other analyses.

### 3.7. Certainty of Evidence Assessment

Studies were also evaluated for their level of certainty of evidence using the GRADE assessment system. Results of the GRADE assessment of meta-analyzed outcomes are presented on Appendix A. Overall, the certainty of the evidence was assessed as weak to very weak.

## 4. Discussion

This is the largest and most comprehensive systematic review and meta-analysis on the effects of preoperative glucocorticoid administration in liver surgery to date. Our results revealed that glucocorticoid prophylaxis did not reduce the overall complication rate in patients undergoing major liver surgery (OR: 0.71, *p* = 0.23), and hence its routine use in this patient population is not supported by sufficient evidence.

Liver surgery presents a unique challenge, being unlike most other major abdominal surgeries in the context of postoperative complications. It has been postulated in the past that underlying the relatively high risk involved in liver surgeries is the cascade of dysfunctional systemic metabolic and hematological responses to injury, which is the result of and also the cause of hepatic dysfunction [51]. When the liver parenchyma is injured, the protective functions of the liver, which would have otherwise compensated for the response to insult, may become impaired or dysregulated [52]. The resulting dysfunction is associated with the typical post-hepatectomy complications such as hepatic insufficiency, bile leakage, wound infections, abdominal infections, pleural effusion, pulmonary atelectasis, and hemorrhage [53]. Liver transplantation follows a similar logic, and the complications may be even more severe [54].

Investigations into the use of glucocorticoids for their hypothesized protective effect against postoperative complications have been ongoing for decades. One of the earliest clinical trials was published in 1996 by Shimada et al. [55]. The authors investigated the effects of steroid administration on postoperative cytokine release and found that a short-term pulse of methylprednisolone might be effective in reducing surgical stress by decreasing cytokine release. Steroids were chosen by researchers for their significant anti-inflammatory effects, which have been hypothesized by trialists as being able to reduce the extent of hepatic dysregulation, allow for a more rapid liver function recovery, and reduce the risk of developing systemic dysregulation in relation to the uncontrolled immune response. However, it should be noted that steroids have long been considered a double-edged sword when it comes to use, as their potential side effects are risky and undesirable [4]. 

Since the two systematic reviews published in 2014, there have been contradictory results published by clinical trialists on the subject of steroids and liver surgery, which necessitated further systematic reviews. While Richardson et al. [12], Li et al. [13], and Yang et al. [14] all reported in their meta-analyses a tendency toward lower overall postoperative complication rates (*p*-values of 0.09, 0.09, and 0.13, respectively), the recent meta-analysis by Hao-Han et al. [15] found a statistically significant decrease in overall complications (*p* = 0.04). However, we cannot validate these results with our updated study. None of the previous meta-analyses were able to detect a statistically significant difference between the intervention and control groups in terms of specific complications, namely, bile leakage, liver failure, wound complications, infectious complications, and pleural effusion. Our analysis of particular complications did not provide a sufficiently high level of evidence, due to the unavailability and/or the improper reporting of these complications. Especially for liver failure, there was an observable difference between the reporting of Onoe et al. [48] and that of other studies. This is possibly due to the different assessments made on what constitutes liver failure. In our meta-analysis, we detected a statistically significant reduction in wound infections, but we have reservations about the quality of the evidence. Firstly, both the sample size and the number of studies are limited, and the intervention groups with zero complications may have introduced a bias toward a reduced odds ratio in the analysis. 

Increased total bilirubin is an indicator of an imbalance between production and excretion and, ultimately, is considered a reflection of liver function [56]. Most of the included studies investigated this outcome as a measure of liver health and, except for Muratore et al. [47], found that steroid administration significantly reduced levels of total bilirubin. Combined with aminotransferases ALT and AST, these are indicators of liver health commonly used in clinical practice. Although the investigation was not as thorough, and the findings were not as consistent as with total bilirubin, there were many significant findings of reduced levels of ALT in the intervention groups. This could signal a liver-protective effect that was bestowed by the intervention, since an increase in ALT is found primarily in the liver and is considered a marker of liver disease [57]. C-reactive protein is an acute-phase protein synthesized by the liver and, along with interleukin 6, is a marker of inflammation. Increases in CRP levels have been associated with liver failure [58]. The studies we reviewed consistently reported significantly reduced levels, signaling a protective effect on the liver. Lastly, prolonged prothrombin time is associated with liver failure [59], as the liver produces many of the factors and components of the coagulation system. Hayashi et al.’s finding [45] on the PTT-INR contradicts other articles included in this review. Coagulation parameters should be considered a critical component of the assessment of liver function; thus, future clinical trials should be designed to generate further high-quality evidence on the effects of the intervention on coagulation.

All previous meta-analyses found significantly reduced levels of total bilirubin in the intervention groups. Although we could not perform a meta-analysis on this outcome, available evidence suggests future clinical trials could validate these findings. All meta-analyses, except for that of Hao-Han et al. [15], have also found significantly reduced levels of IL-6 in the intervention groups. However, IL-6 has not been measured in recent clinical trials. We recommend that IL-6 be included in future clinical trial designs as an outcome measure.

The reporting of laboratory measurements as outcomes was not always consistent across the included studies. Although we did not detect a considerable risk of bias using Cochrane’s tools, in the clinical context, it might have been more useful to have explicitly detailed and consistent measurements taken throughout the follow-up period. Furthermore, the mathematical analysis of the aggregated data should always be presented in the publication along with distribution, in order to enable reliable meta-analyses. Laboratory outcomes should be examined and reported in a way that is consistent with complications and patient subgroups. Peak values should also be examined alongside means, and the measurements should be documented clearly with their time points to reduce the risk of bias. Measurement results and time points left out of the reports without a clear explanation presented a challenge in conducting our meta-analysis. Another challenge was results reported without reliable distribution figures, which made meta-analyzing these outcomes by pooling medians and means unreliable.

We recommend that trialists design future randomized clinical trials around an internationally acknowledged postoperative complication classification system such as the Clavien–Dindo Classification System [60] or the Comprehensive Complication Index (CCI) [18], which is an integrated complication-reporting algorithm. 

On the other hand, we recommend that future clinical trials put emphasis on differentiating the benefits for patient subgroups, categorized according to the indication for liver surgery, as well as patient severity scoring systems. We recommend utilizing the APACHE IV scoring system [61] for assessing critically ill patients, and the American Society of Anesthesiologists (ASA) physical status classification system [62] to group patients according to the assessed surgical risk. Furthermore, trialists ought to consider the potential difference in benefits derived for patients undergoing liver transplantation versus open or laparoscopic hepatic resections. Researchers may be able to detect differences in benefits derived between different regimes of preoperative steroid administration. Therefore, designing future clinical trials around contrasting single high-dose preoperative administration versus progressively decreasing doses of perioperative administration on subsequent days, as designed by Onoe S. et al. [48], might yield a higher level of evidence.

The ERAS Society’s recommendation on perioperative steroid administration in their 2016 guideline [11] is currently stated as a weak recommendation based on a moderate level of evidence. In light of our systematic review and other studies that have been published since 2016, we recommend that the guidelines on this intervention be updated with new levels of evidence and a new grade of recommendation. 

### 4.1. Strengths and Limitations

Our study had certain strengths and limitations that should illuminate clinical decision making and future clinical trial designs. Our study included the most recent publications on the topic and had considerably more patients in the analysis compared to the previous meta-analyses. All included articles were randomized controlled trials which were critically appraised using the GRADE approach to the level of evidence certainty, which was missing from the literature. As such, the qualitative assessment within this manuscript describes where there is uncertainty in the currently available literature. 

Our study was limited by data availability, which prevented us from performing subgroup analyses, and meta-analyses on postoperative laboratory outcomes. Differences in intervention regimes may limit the generalizability of our findings. Furthermore, the analyses were limited by the considerable heterogeneity between studies, which limited the applicability of our findings. Finally, we could not perform an assessment of publication bias due to the low number of studies.

### 4.2. Implications for Practice and Research

We were unable to show any convincing benefits to using glucocorticoids preoperatively in liver surgery, and hence the routine use of preoperative glucocorticoids in major liver surgery cannot be supported by evidence. However, it should be noted that there were no reported cases of adverse events associated with its use either. Therefore, its use should only be warranted within the domain of clinical research. 

Further prospective data collection is needed to assess the benefits of perioperative steroid administration on particular postoperative complications. Mainly, the effects on liver dysfunction or failure, shock, septic complications, and coagulation-related complications should be investigated.

It is crucially important to bring scientific results to the bedside [63,64]. As such, research on this particular topic should focus on outcomes that are specific to patient populations and direct clinical outcomes with rigorous postoperative follow-ups. 

## 5. Conclusions

In conclusion, our meta-analysis did not show any statistically significant reduction in postoperative complications for patients undergoing liver surgery, except for in the rate of developing wound infections. However, further investigation is needed to clarify this finding. Most clinical trials reported significant improvements in postoperative laboratory values at different time points, which signifies a protective effect against liver injury and dysfunction, but further research is needed for a higher grade of evidence.

## Figures and Tables

**Figure 1 jcm-13-02097-f001:**
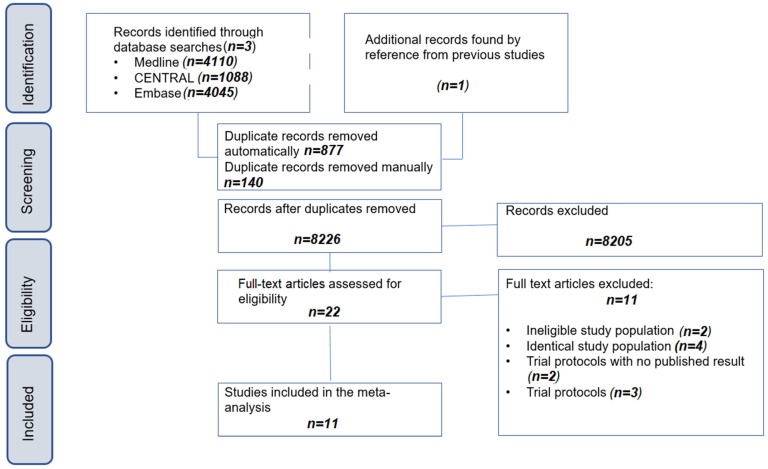
PRISMA flowchart of selection describing the systematic search and selection process.

**Figure 2 jcm-13-02097-f002:**
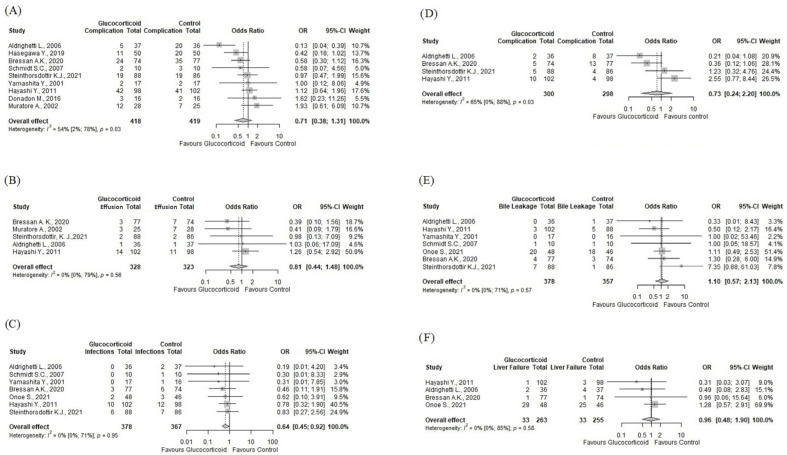
Forest plots of clinical outcomes. (**A**) overall postoperative complication rate; (**B**) pleural effusion; (**C**) wound infection; (**D**) septic/infectious complications; (**E**) bile leakage; (**F**) liver failure of any grade [40,41,42,43,44,45,46,47,48,49]. OR: odds ratio; CI: confidence interval.

**Figure 3 jcm-13-02097-f003:**
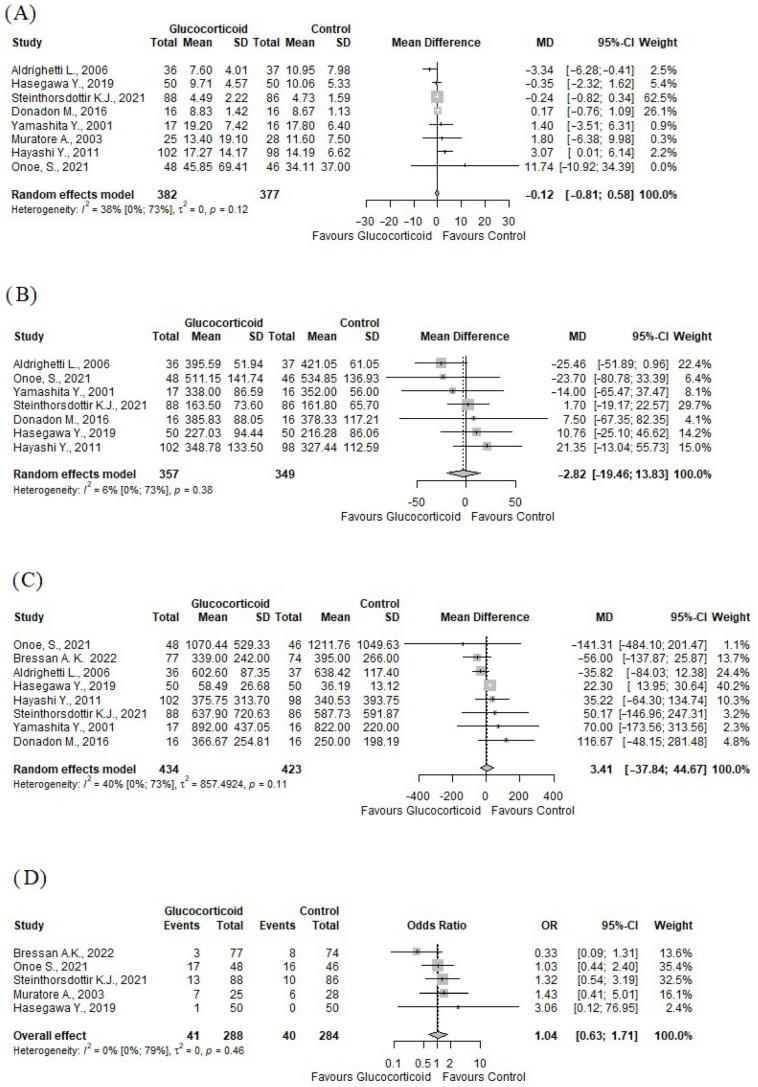
Forest plots of other outcomes. (**A**) length of hospital stay; (**B**) total operative time; (**C**) blood loss (milliliters); (**D**) need for administration of blood products [40,41,42,43,44,45,46,47,48]. OR: odds ratio; CI: confidence interval; MD: mean difference; SD: standard deviation.

**Figure 4 jcm-13-02097-f004:**
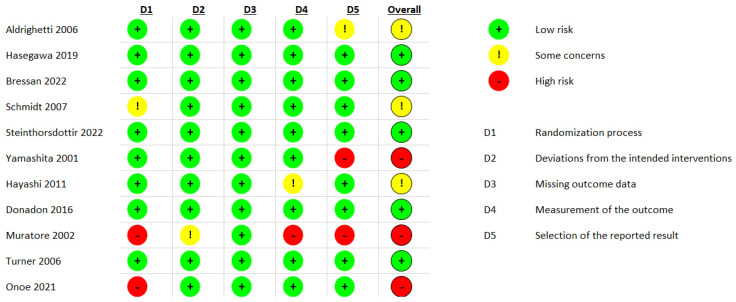
Results of the risk of bias assessments using RoB2 [40,41,42,43,44,45,46,47,48,49,50].

**Table 1 jcm-13-02097-t001:** The summary of the studies included (author, publication date, country, patient distribution, and demographic data).

First Author and Publication Date	Intervention	Control	Surgery Type	Patient Distribution		Age, Years		Sex, Female % of Total	
				Intervention	Control	Intervention	Control	Intervention	Control
Aldrighetti L. 2006 [40]	IV Methylprednisolone 500 mg	Unclear	Hepatic resection	36	37	61.8 (21–78) ^c^	63 (31–85) ^c^	37.83	38.88
Steinthorsdottir K. J. 2021 [41]	IV Methylprednisolone 10 mg/kg	Standard of care including IV Dexamethasone 8 mg	Open liver surgery without biliary reconstruction	86	88	65.2 ± 11.2 ^b^	64.4 ± 12.0 ^b^	34	30.6
Bressan A. K. 2022 [42]	IV Methylprednisolone 500 mg	Placebo	Hepatic resection	74	77	63.9 ^a^	62.4 ^a^	47.2	38.9
Hasegawa Y. 2019 [43]	IV Methylprednisolone 500 mg	Placebo	Hepatic resection	50	50	67 (59–74) ^c^	68 (62–75) ^c^	38	40
Donadon M. 2016 [44]	IV Methylprednisolone 500 mg	Placebo	Hepatic resection	16	16	65 (27–80) ^c^	63 (22–77) ^c^	44	37.5
Hayashi Y. 2011 [45]	IV Hydrocortisone 500-300-100 mg consecutively	Non-administration	Hepatic resection	98	102	69 (39–81) ^c^	70 (35–82) ^c^	No data	No data
Yamashita Y. 2001 [46]	IV Methylprednisolone 500 mg	Non-administration	Hepatic resection	16	17	56.8 ^a^	60.3 ^a^	31.25	23.52
Muratore A. 2002 [47]	IV Methylprednisolone 30 mg/kg	Non-administration	Hepatic resection	28	25	64.1 ^a^	65.4 ^a^	60.7	32
Onoe S. 2021 [48]	IV Hydrocortisone 500-300-200-100 mg	Placebo	Combined liver and extrahepatic bile duct resection	46	48	70 (39–83) ^c^	71 (39–84) ^c^	33	40
Schmidt S. C. 2007 [49]	Methylprednisolone 30 mg/kg	Placebo	Hepatic resection	10	10	65 ^a^	57 ^a^	60	70
Turner S. 2006 [50]	IV Methylprednisolone 10 mg/kg	Placebo	Orthotopic liver transplantation	17	17	53.4 ^a^	57.7 ^a^	35.3	35.3

RCT: randomized controlled trial, ^a^ = mean, ^b^ = mean ± standard deviation, ^c^ = median (range).

## Data Availability

The datasets used in this study can be found in the full-text articles included in the systematic review and meta-analysis.

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
