# Peer review of "The Effect of Preoperative Administration of Glucocorticoids on the Postoperative Complication Rate in Liver Surgery: A Systematic Review and Meta-Analysis of Randomized Controlled Trials"

_jcm, 2024, doi:10.3390/jcm13072097_

Round 1
Reviewer 1 Report
Comments and Suggestions for Authors
With pleasure, I read the paper titled “The Effect of Preoperative Administration of Glucocorticoids on the Postoperative Complication Rate in Liver Surgery: A Systematic Review and Meta-analysis of Randomized Controlled Trials”. The topic is clinically relevant to practice, and of importance to the readers of the Journal of Clinical Medicine. Overall, the manuscript reads well and has good flow of ideas, relevant citations, and good summary of data using tables and figures. The introduction section was detailed enough to provide the reader with the needful background information; however minor edits are needed to highlight the significance of the study in addition to the need to conclude the section with some hypotheses. The methods section was detailed too, however, some MAJOR and MINOR edits are needed for complete reporting and compliance with PRISAM. Again, the meta-analysis of various endpoints is a key strength of this manuscript. Nonetheless, some attention is needed for specific parts of the manuscript and also the results of publication bias should be omitted as described below. The discussion section provided some elaboration and comparison with previously published literature; however, the significance of the study is ought to be clearly pinpointed. Attention is to be paid regarding the GRADE certainty of evidence as it could be seriously misleading. The research had some unavoidable limitations, all of which had been explicitly acknowledged, but additional limitations should be elaborated. The conclusion is line with the presented results. ALL IN ALL, this manuscript is clinically relevant. The manuscript is not suitable for publication in its current form and a Major revision is required:
ABSTRACT
All abbreviations must be spelled out upon first encounter. Please mention the dates of search from when to when. For each reported outcome, please mention the number of RCTs. Pleas provide a conclusion.
INTRODUCTION
(a) Please highlight the significant aspects of your study. Is it the most up-to-date meta-analysis? Does it include more RCTs? Does it examine further aspects (such as publication bias, leave-one-out sensitivity analysis, trial sequential analysis, or GRADE quality of evidence) compared with the previous meta-analysis articles?
(b) Please conclude the introduction section with some proposed hypotheses.
METHODS
(a) The use of “no restrictions” is vague. Please mention if specific filters (such as year of research, country of publication, or English language) were used during literature screening.
(b) Have you searched the grey literature or the reference lists of the included RCTs for additional studies that could have been missed?
(c) I believe the date of search may be outdated (April 1, 2023); almost 1 year ago. The authors are encouraged to update their literature search as additional RCTs could have been missed in their analysis.
(d) Please specify the type of the control. Is it “placebo”, “no intervention”, or both?
(e) It is recommended to report the inclusion criteria using the standard evidence based PICOS method.
(f) Please indicate why the random-effects model was used. Was heterogeneity expected a priori?
(g) Why did you choose odds ratio over risk ratio for evaluation of dichotomous outcomes?
(h) For testing between-study heterogeneity, what were the cut off of establishing significant heterogeneity based on the Higgins I2 value and the p-value (<0.1) of the chi-square Cochran's Q statistic?
(i) Have the authors performed leave-one-out sensitivity analysis or trial sequential analysis?
RESULTS
(a) For Table 1 and Suppl Table 1, several characteristics are missing. Please specify the type of surgery (liver resection or transplantation), type of control (placebo or no intervention), and details of glucocorticoids (mode of administration: oral or IV). All studies are RCTs (as indicated in your review) and this column may be removed.
(b) For Figure 2, it is very difficult to visualize. I recommend splitting the figure into two figures and make them larger.
(c) For Figure 4, I would encourage the authors to revisit their ROB analysis and make sure it was properly performed.
(d) The data on certainty of evidence should be rechecked again. In fact, the authors need to provide details for their judgement for all domains of the GRADE pertaining to risk of bias, indirectness, imprecision, inconsistency, publication bias, etc. Considering some outcomes had relatively small number of studies, small number of sample size, high heterogeneity, wide confidence intervals, and no way or reliably assessing publication, then the certainty of evidence should be downgraded. The “high” certainty of evidence means that the investigators are very confident that the effect they found across studies is close to the true effect and further research is very unlikely to change our confidence in the estimate of effect. This contradicts with the authors’ call for prospective RCTs to be conducted.
(e) The data on publication bias should be omitted. This is because according to Egger et al, publication bias evaluation using funnel plots and quantitative testing is not reliable or powered enough and could be very misleading when the number of studies is less than a minimum of 10.
(f) You may want to strengthen the robustness of the results by performing leave-one-out sensitivity analysis and/or trial sequential analysis.
DISCUSSION
(a) Please emphasize the significance of your study findings and how your study differs from the previous meta-analyses. Please refer to above comment in the Introduction section (a).
(b) Please acknowledge additional limitations, such as: (a) some outcomes had high heterogeneity which could be ascribed to differences in study durations, patient characteristics, and control groups, and (b) publication bias was not evaluated due to the small number of included studies (<10).
OVERALL
(a) The manuscript needs minor polishing for English language and editing.
(b) For an additional line of validation/accuracy, in view of the large number of the presented figures, please double-check again that results are matched between figures and data in the results section.
Comments on the Quality of English LanguageMinor english editing
Reviewer 2 Report
Comments and Suggestions for Authors
This is a very interesting meta-analysis concentrated on the effect of preoperative administration of glucocorticoids on the postoperative complication rate in liver surgery. I have several suggestions for this study.
1, In 2023, a nearly same meta-analysis titled " Short-term outcomes of perioperative glucocorticoid administration in patients undergoing liver surgery: a systematic review and meta-analysis of randomised controlled trials."(PUBMEDID:37169506). However, the included article and the results of this article are different with yours.
2, Concentrated indicators for postoperative complication is not enough and very unclear. For exmple, postoperative complication rate is mentioned as a indicator. However, what is the defination of postoperative complication is not mentioned.
3, Postoperative complication is very complicated. It is correlated with many other factors including operation methods (open/ laprascopic), post operation drug administration, resection scope (1-3 section of liver). Mabye it could not be explained by a single study. But some of such factors should be considered as a bias.
4, Picture size and clarity is not the same for figure2 and 3.
Comments on the Quality of English Language
Minor editing of English language required
Reviewer 3 Report
Comments and Suggestions for Authors
The article investigates the impact of preoperative glucocorticoids administration on postoperative complication rates in liver surgery, through a systematic review and meta-analysis of randomized controlled trials. The findings suggest a non-significant reduction in overall postoperative complications with glucocorticoid use, highlighting the need for further research in this area.
I congratulate the authors for such a fine detailed study methodology, the transparence of reporting, the accurate utilization of the statistical tools, with citations! It was a pleasure to read the paper!
Minor issues are only to be resolved.
Methods
It's important to clarify that PRISMA is a reporting guideline rather than a methodological guideline for performing reviews. Please change the wording to we reported our systematic review ... in accordance to PRISMA, and we undertook our research based on reccomendation of the Cochrane ...
As you correctly postulated, having less then 10 studies precludes publication bias assesment, thus the text regarding funnel plots ... should be removed.
Also, no drapery plots, nor prediction intervals were reported in the article. Thus, these statements should be erased.
Results
The manuscript correctly employs the RoB 2 tool for assessing the risk of bias in included randomized controlled trials, adhering to current best practices recommended by Cochrane.
However it fells the assessment of the RoB should be reenacted. I looked at the study Muratore 2002, and at first glance I could not observe they use allocation concealment, nor the method of randomization was not specified. Even if the groups were balanced in their caracteristics, RoB would put the study domain at some risk, while in your assessment is low risk. Maybe I overlook something. I verified the first 5 studies for this domain and I found no other problems.
For figures, please add the abbreviations for OR, CI, MD, SD.
Round 2
Reviewer 1 Report
Comments and Suggestions for Authors
The authors did a great job by addressing most of the comments to their best. The manuscript needs read well and is more scientifically sound and methodologically robust. Despite the leave-one-out sensitivity analysis did not show major difference, however, it would have been great to include them as supplementary. The GRADE data are realistically better now.
Comments on the Quality of English Languageroutine English editing during copyediting.